

Characterization of three new condensation particle counters for sub-3 nm particle detection: ADI
versatile water CPC, TSI 3777 nano enhancer and boosted TSI 3010
Juha Kangasluoma[1], Susanne Hering[2], David Picard[3], Greg Lewis[2], Joonas Enroth[1], Frans Korhonen[1],
Markku Kulmala[1], Karine Sellegri[3], Michel Attoui[1,4], Tuukka Petäjä[1]
[1] Department of Physics, P.O. Box 64, 00014, University of Helsinki, Helsinki, Finland
[2] Aerosol Dynamics Inc., Berkeley, CA, USA
[3] Laboratoire de Météorologie Physique, UMR6016, Observatoire de Physique du Globe de
Clermont-Ferrand, CNRS – Université Blaise Pascal, Clermont-Ferrand, France
[4] University Paris Est Creteil, University Paris-Diderot, LISA, UMR CNRS 7583, France
Abstract
The scientific need to understand nanoparticle dynamics at sizes below 3 nm has pushed companies
to develop commercial solutions to measure particles down to 1 nm. In this study we characterize the
performance of three new particle counters able to detect particles smaller than 3 nm: Aerosol
Dynamics Inc versatile water condensation particle counter (v-WCPC, ADI, Berkeley, USA), TSI 3777
nano enhancer (TSI Inc., Shoreview, USA) and modified and boosted 3010 type CPC from Clermont
Ferrand University called as B3010. The 3777 and v-WCPC were characterized using tungsten oxide
test particles with all charging states: negative, positive and neutral, and with positively charged
tetradodecylammonium bromide. The detection efficiencies of the particle counters were measured
with two different temperature settings: low temperature difference settings so that the CPCs did not
detect any ions from a radioactive source; and high temperature difference settings so that the
supersaturation was at the onset of homogeneous nucleation for the 3777, or confined within the
range of liquid water for the ADI v-WCPC. The measured 50% detection diameters (d50) were in the
range of 1.3 – 2.4 nm for the tungsten oxide particles depending on the particle charging state and
CPC temperature settings, and between 2.5 and 3.3 nm for the organic test aerosol for the 3777 and
v-WCPC. The d50s were measured for the B3010 with negatively charged tungsten oxide particles with
four different inlet flow rates. The v-WCPC and 3777 were also compared side by side by measuring
atmospheric aerosol, exhibiting an excellent agreement.
1 Introduction
The work of Stolzenburg and McMurry (1991) started a new chapter in aerosol research with their
prototype laminar flow condensation particle counter (CPC) capable of detecting 3 nm particles via
condensation of butanol vapor. The significant improvements in the instrument included minimized
diffusion losses in the sampling line and a sheath flow in the condenser to focus the particle beam in
the maximum butanol supersaturation in the middle of the condenser (Wilson et al., 1983). This
instrument is the predecessor of the ultrafine CPC 3025A and 3776 (TSI Inc., Shoreview, USA), which
currently are widely used in various fields of aerosol science to study particle dynamics at particle sizes
larger than 3 nm (e.g. Aalto et al., 2001; Weber et al., 1996).
43          It was not possible to detect particles smaller than 3 nm with the CPC technology until 1997,
when Seto et al. (1997) published their design on the particle size magnifier (PSM) used to study
heterogeneous nucleation of dibutyl phthalate vapor onto small ions. Their advances were made
possible by the development of a new differential mobility analyzer (DMA) combined to an
electrospray source, allowing the testing of the CPC with well-characterized monomobile samples. The
CPC itself was based on the design of Okuyama et al. (1984), which is a mixing type CPC. It took until
2011 to commercialize the mixing type CPC technology, when Vanhanen et al. (2011) published their
version of the diethylene glycol (DEG) based PSM, today sold as the Airmodus A10 PSM (A11 nano
condensation nuclei counter when combined to Airmodus A20 butanol CPC).



The first use of DEG as a working fluid was by Iida et al. (2009), who studied sub-3 nm particle
detection with various different working fluids theoretically and experimentally. They modified the
TSI 3025A to operate with DEG and showed particle activation and growth down to 1 nm. Because the
DEG droplets formed are small, a traditional, butanol based CPC is used as a detector. The idea of
using the commercial TSI instrument with modifications to operate with DEG has been followed by
several other researchers (Jiang et al., 2011a; Jiang et al., 2011b; Kuang et al., 2012a; Kuang et al.,
2012b; Wimmer et al., 2013). In 2016 TSI commercialized a DEG instrument based on the work of Iida
et al. (2009). This instrument, the TSI 3777 nano enhancer (3777), is one of the three instruments
characterized in this study.
Generally, laminar flow ultrafine CPCs use a sheathed condenser, which makes the CPC design
more complex compared to non-sheathed CPCs. Yet recent efforts have shown lower detection limits
with unsheathed laminar-flow instruments. Particle detection with the butanol-based TSI 3010 has
been shown down to 2.5 nm from the factory settings d50 (diameter at which 50% of sampled particles
are detected) of 10 nm (Mertes et al., 1995; Russell et al., 1996; Wiedensohler et al., 1997).
Kangasluoma et al. (2015a) showed 1 nm particle detection with the commercial unsheathed
condenser CPCs TSI 3772 and Airmodus A20 by increasing the temperature difference between the
saturator and condenser up to 40 °C. The second CPC characterized in this study is a boosted TSI 3010
(B3010), which is a modification of the commercial TSI3010 developed at the Université Blaise Pascal
in which the temperature control of the saturator and condenser is decoupled to allow free selection
of the temperatures, and critical orifice is replaced with a flow meter and a miniature rotary vane
pump.
The disadvantage of all the previous CPCs is the slight toxicity of the working fluids butanol
and DEG. Hering et al. (2005) addressed this issue by developing a water based, laminar technology
(Hering and Stolzenburg, 2005), which was commercialized as the TSI WCPC models 3785 (Hering et
al., 2005) and 3786 (Ida et al, 2008), and subsequently as the 3783, 3787 and 3788. In the Model 3786,
and later in the Model 3788 (Kupc et al., 2013), small particle detection was enabled by introducing
similar sheathed condenser as in the butanol based CPCs
The ADI versatile WCPC (v-WCPC), which is the third CPC characterized in this study, advances
the laminar-flow water-based CPC through a three stage design that reduces the water vapor
concentration and temperature in the growth tube after the peak supersaturation is achieved, and
yet allows for continued particle growth (Hering et al., 2016). This three-stage approach facilitates
higher temperature differences between the first two stages, and can produce higher peak
supersaturation values than the ultrafine TSI 3786 or TSI 3788. The v-WCPC is an unsheathed
instrument, operating at an aerosol flow of 0.3 litres per minute (lpm) and at more extreme
temperatures than all of the current commercial TSI WCPCs. In contrast to the DEG-based instruments,
which require a separate CPC as a detector due to the small size of the DEG droplets, the droplets
formed in the growth tube of the v-WCPC are sufficiently large to be detected directly.
The aim of this study is to characterize the performance of the v-WCPC and 3777 with two
different types of test particles, to measure effect of charge to the detection efficiency and to compare
the instrument responses in atmospheric sampling. The performance of the B3010 was characterized
with tungsten oxide particles for different aerosol flow rates.
2 Experimental
2.1 Condensation Particle Counters
A flow diagram of the 3777 is presented in Figure 1. The design is largely similar to the TSI ultrafine
3776. The inlet flow rate is 2.5 lpm, of that 1.5 lpm being transport flow and 1 lpm split as the sheath
flow (0.85 lpm) and aerosol flow (0.15 pm). The sheath flow passes through a dessicant drier to
remove most water vapor entering the condenser, possibly altering the detection efficiency of a DEG
based CPC (Iida et al., 2009; Kangasluoma et al., 2013). Downstream of the drier the sheath flow is





saturated with DEG before entering the condenser around the aerosol flow, which is guided in the
centre line of the condenser. The 3777 does not have its own optics head, as the droplets formed by
DEG condensation are too small for direct detection. Instead the detector is a TSI 3772 CPC, which
further enlarges and then counts the droplets pre-grown by DEG in the condenser. The factory settings
of the 3777 are: saturator 62 °C and condenser 12 °C (low temperature difference (dT) settings). At
these settings the 3777 did not detect any ions produced by a radioactive [241]Am source. It was also
operated at boosted settings so that the supersaturation was at the onset of homogeneous
nucleation. With the boosted settings the saturator was 70 °C and condenser 7 °C (high dT settings).
111        Flow diagram of the ADI v-WCPC is presented in Figure 2. The v-WCPC does not require a
separate CPC for droplet detection, nor does it use a sheath flow, making it a relatively simple CPC.
The v-WCPC has two flows, a transport flow and an aerosol flow, both of which are controlled by
critical orifices. For experiments conducted here the inlet flow rate of the v-WCPC was 2.2 lpm, of
which 1.9 lpm is transport flow and 0.3 lpm aerosol flow. The aerosol flow passes upward through a
three-stage growth tube consisting of a cool-walled conditioner, followed by a short, warm-walled
initiator, and subsequently followed by a cool-walled moderator (Hering et al., 2014). A continuous
wick spans all three growth tube sections.  Liquid water is injected at a rate of 1 µL/min at the initiator,
and excess drains toward the inlet and is removed with the transport flow. Peak supersaturation and
particle activation occurs within the initiator, and growth continues in the moderator. The formed
droplets are counted by an optics head mounted directly at the outlet of the growth tube. Further
detail is presented by Hering et al. (2016). The v-WCPC was tested at two different temperature
settings: conditioner at 8 °C and initiator at 90 °C (low dT settings), corresponding to supersaturation
low enough to not detect any ions from a radioactive source, and boosted settings with conditioner
at 1 °C and initiator at 95 °C (high dT settings), which is close to the extremes attainable without
freezing or boiling. In both instances the moderator was operated at 22°C, and the optics head at 40°C.
127        The B3010 is based on a cheap second hand TSI 3010, from which everything except the
saturator block, the condenser and the optical detector are removed. The original electronics have
been completely replaced with custom made boards, to handle the higher power consumption, and
operate off 28 VDC, the primary power supply in aircrafts. The whole system is controlled by a credit
card sized ARM computer, running a tailor-made embedded Linux operating system. It features a
touchscreen, a TSI-like serial port protocol, and TTL pulse output. With these modifications the
saturator heating and condenser cooling are decoupled. In addition, the critical orifice and external
heavy pump are replaced by a laminar flowmeter and a miniature rotary vane pump. The user may set
the temperature of the saturator, condenser and optics as well as the flow rate, independently from
one another. The B3010 was operated at saturator temperature 55 °C, optics head 56 °C and condenser
11 °C. The B3010 will be described in more details in a dedicated article, presently in preparation. Table
1 summarizes the instrument operation conditions.
2.2 Aerosol generation
Two methods were used to generate the test aerosol: glowing wire generator (GWG) and electrospray
source. In the GWG (Peineke et al., 2006), a thin, 0.4 mm in diameter, tungsten wire is heated
resistively in a metal chamber. The wire is flushed with 5.0 $N_2$ flow and it has been shown that
negatively charged tungsten oxide clusters are formed into the $N_2$ flow without additional charging
(Kangasluoma et al., 2015b). Positively charged clusters contain some hydrocarbon molecules
clustered with tungsten oxide, explaining why usually the measured d50 usually is larger for positively
than negatively charged clusters (Kangasluoma et al., 2016b). The d50 for all three CPCs, at two
different temperature settings for the 3777 and v-WCPC and at one settings for the B3010, was
measured with tungsten oxide particles by size selecting 18 different sizes of particles between 1 and
4.5 nm with the Herrmann type high resolution DMA (Kangasluoma et al., 2016a) (Figure 4). The tubing
lengths downstream of the DMA were selected to be equal for the vWCPC, 3777 and electrometer so
that the particle penetration through the tubes can be considered equal. For the B3010 the line length



was approximately half of the electrometer tubing for the same reason. The d50 of the 3777 was
measured at four different sample flow dew points with negatively charged tungsten oxide particles.
Water vapor was added to the sample flow with a humidified dilution flow downstream of the DMA.
The d50 for the B3010 was measured also at four different aerosol flow rates, 0.5, 1.0, 1.4 and 1.6
lpm, by varying the rotary vane pump speed.
The electrospray source produces charged sample molecule containing droplets by spraying
liquid at high voltage out of a capillary needle against a grounded electrode. The charged droplets are
close to the Rayleigh limit, and produce charged sample molecules and clusters to the gas flow by
series of Coulomb explosions, and ion and solvent evaporation from the droplet. The highly charged
droplets can be close to 2 nm in mobility diameter (Ude and Fernández de la Mora, 2005), for which
we neutralized the flow exiting the electrospray with a radioactive [241]Am source to also be able to
sample the clusters larger than 2 nm (Kangasluoma et al., 2016a). The electrosprayed sample in the
experiments was tetradodecylammonium bromide (TDDABr) (Ude and Fernández de la Mora, 2005).
d50 was measured for the 3777 and v-WCPC with TDDABr with the low dT settings, and for the v-
WCPC at the high dT settings. For the 3777 we could not measure the d50 at high dT settings due to
the fact that the aerosol-to-sheath flow ratio is very sensitive to the CPC inlet pressure, and TDDABr
was produced by drawing the flow out of the DMA, leading to a pressure drop of approximately 5 kPa
at the CPC inlets. This pressure drop was enough to alter the aerosol-to-sheath flow ratio in the 3777
and cause homogeneous droplet formation at high dT settings.
To measure the d50 for neutral particles, we followed the approach presented in our previous
studies (Kangasluoma et al., 2015b; Kangasluoma et al., 2016b). The sample flow downstream of the
DMA passes through a mixing chamber, to which a tube containing a [241]Am radioactive is connected.
0.2 lpm of the sample flow is drawn through the tube, and ions from the radioactive source are drifted
to the mixing chamber against the counter flow with an electric field. A fraction of the sample particles
are neutralized by the opposite polarity ions drifted to the mixing chamber. An ion precipitator is
placed downstream of the mixing chamber to allow sampling of only neutral particles with the CPC.
The concentration detected with the CPC is normalized against the electrometer. The detection
efficiency curve is further normalized with detection efficiency of the largest selected diameters where
the role of charge on the detection efficiency is assumed to be negligible. This method yields
uncertainties in the resulting d50 due to possibly size dependent neutralization efficiency and
chemical composition of the neutralized particles, however, it is being the only method to measure
d50 for neutral particles for sub-3 nm particles. Neutral d50 was measured for both instruments with
high and low dT by neutralizing both negatively and positively charged particles.
2.3 Ambient sampling setups
The response of the v-WCPC was measured against the TSI electrometer (model 3068B) at different
concentrations at sizes 1.4nm, 1.8nm, 2.4nm and 4.4 nm. The concentration at each size was
controlled by adding a dilution flow of compressed and filtered air downstream of the DMA.
Simultaneous data were collected for the 3777, however the dilution flow was enough to change the
aerosol-to-sheath flow ratio of the 3777 due to a small change in the inlet pressure, and therefore the
3777 data of this experiment are not presented. However, assuming that any possible undercounting
at high concentration originates from particle coincidence in the optics, the concentration calibration
of the 3777 should be practically the same as of the 3772 CPC when the dilution of 0.15/1 is taken into
account.
Finally, the 3777 and v-WCPC were placed to sample atmospheric aerosol from Helsinki city
area. The instruments were sampling from the same inlet for approximately 18 h to compare the
measured concentrations from atmospheric aerosol. The v-WCPC data were dead-time corrected
using the dead time correction factor derived from the concentration dependent response for 4.4 nm.
3 Results



3.1 Detection efficiency
Figure 5 presents the d50 measurements for the B3010, 3777 and v-WCPC at low dT settings for
positively and negatively charged tungsten oxide particles. The standard deviation in the detection
efficiency data was in most cases < 5%, which is why it is not plotted in the figures. X-axis uncertainty
can be taken from the Herrmann DMA resolution of approximately 20, which leads to relative
uncertainty of ± 5% based on the selected mobility peak full width at half maximum of 5%. Therefore,
uncertainties in the data arise mostly from other sources, such as unequal sampling line penetration
or possibly changing particle chemical composition as a function of size. At these settings none of the
CPCs detect the ions generated by a bipolar ion source, such as is commonly used for mobility based
particle size distribution measurements. We find that the v-WCPC exhibits slightly lower d50 than the
3777, while the d50 of the B3010 is clearly the highest. The d50 of 3.2 – 3.4 nm for the B3010 however
shows that the conventional TSI 3010 can be boosted to similar performance as the TSI ultrafine 3776,
just with a shallower d50 curve due to larger particle diffusion losses, by decoupling the heating and
cooling of the saturator and condenser. Respective d50 values for the B3010, v-WCPC and TSI-3777
are 3.4 nm, 1.7 nm and 1.8 nm for negatively charged tungsten oxide, and 3.2 nm, 1.9 nm and 2.0 nm
for positively charged.
223          At high dT settings the d50 curves are presented in Figure 6. For the 3777 the temperatures
were selected as those that are just below the limit of homogeneous nucleation of the DEG working
fluid.  For the v-WCPC, the temperatures are simply the largest extremes attainable without freezing
or boiling the water working fluid.  Unlike the DEG instrument, the high dT operation of the v-WCPC
is not near the homogeneous nucleation limit, as no evidence of homogeneous nucleation was
observed even at reduced inlet pressures. At these higher dT settings, we find somewhat more
efficient detection of smaller particles by the 3777 than the v-WCPC. The d50s are lowered to 1.4 nm
and 1.3 nm for negatively charged, and to 1.5 nm and 1.4 nm for positively charged for the v-WCPC
and 3777, respectively.
232          Table 2 summarizes the measured d50 for all experiments. The d50 for 3777 and v-WCPC at
both settings is lower for negatively charged particles than for positively charged particles. This is
observed throughout the past literature (Kangasluoma et al., 2014; Kuang et al., 2012b; Sipilä et al.,
2009; Stolzenburg and McMurry, 1991; Winkler et al., 2008), and explained by hydrocarbon
contaminants in the positively charged particles (Kangasluoma et al., 2016b). Based on previous
literature (Kangasluoma et al., 2014; Kuang et al., 2012b) slightly lower d50 values can be expected
for inorganic salt particles than the measured d50s for tungsten oxide particles in this study. TSI states
in their instrument brochure a d50 of 1.4 nm for negatively charged NaCl particles at factory settings
(low dT in this study), which is well in line with this study. Similarly, the d50 values reported here for
the v-WCPC are close to those observed by Hering et al. (2016) who measured d50 of 1.6 nm and
1.9 nm for high dT and low dT operation, respectively, for particles from a heated NiCr wire.
243          Figure 7 presents the d50 curves for the neutralized tungsten oxide particles. The data is
normalized so that the mean of 3 largest diameters is 90% based on the assumption that at those sizes
the charge does not affect the detection efficiency anymore. Also is assumed, through the
normalization that the neutralization efficiency does not change as a function of the particle size.
Further uncertainties arise from the unknown processes that take place during neutralization. Due to
these uncertainties, the curves are not as smooth as for the charged particles. However, an estimate
for the neutral d50 will be obtained from these experiments, which are 1.6 nm and 1.5 nm for v-WCPC
and 3777 at high dT, and 2.0 nm and 1.9 nm at low dT settings, respectively, for neutralized negatively
charged tungsten oxide. For positively charged particles the respective values are 2.2 nm and 2.1 nm
at high dT settings and 2.4 nm and 2.3 nm at low dT settings (Figure 9). The neutral d50s are greater
than for charged d50 values by approximately 0.1-0.5 nm at low dT settings, similar to that obtained
in Kangasluoma et al. (2016b) for water-tungsten oxide and DEG-tungsten oxide system.
The d50 curves for positively charged TDDABr for the 3777 and v-WCPC are presented in
Figure 9. For both instruments the d50 values are higher than for tungsten oxide particles, but this is
most pronounced for the v-WCPC.  At the low dT settings d50 values are 3.3 nm and 2.5 nm at for the
v-WCPC and 3777 respectively, and 2.8 nm for the v-WCPC at high dT.  At the high dT and the reduced
inlet pressure for these TDDABr tests, the 3777 produced homogeneously nucleated particles, and
hence its high dT d50 value could not be measured. These differences in the d50 imply that the CPCs
should be calibrated with the same aerosol composition as with the real experiment is conducted.
3.2 Effect of sample dew point for the 3777
Because water-contamination was observed as a source of error in the original laminar-flow DEG
instruments, this question was examined for the 3777. The response of the 3777 for negatively
charged tungsten oxide particles as a function of sample flow dew point is presented in Figure 10. The
observed variation with dew points ranging from completely dry gas to 20 °C in the d50 is only
approximately 0.1 nm. The apparently increased plateau value for the highest dew point can be due
to slightly higher inlet pressure, increasing the aerosol flow of the instrument. The variation in the d50
due to changing dew point is less than compared to for example 0.3 nm reported in Kangasluoma et
al. (2013) for the Airmodus A09 PSM. This is due to the smaller amount of sample flow water vapor
reaching the condenser in the 3777, since 85% of the condenser flow is dried, as compared to 0% of
the condenser flow of the PSM of that time. At that time the PSM used internal pumps, today they are
replaced by mass flow controllers which are usually fed by dry compressed air, resulting to dried
condenser flow fraction from 4% to 34% depending on the instrument operation.
3.3 Effect of flow rate on the B3010
Results from the inlet flow rate experiment for the B3010 is presented in Figure 11. The d50 curve at
aerosol flow rates of 0.5, 1, 1.4 and 1.6 lpm are rather similar within the experimental uncertainties,
while at flow rate of 0.5 the detection efficiency clearly deviates to lower values at particle diameters
larger than 3 nm. This can be possibly due to larger final droplet diameters and subsequent
gravitational losses at the low flow rate. Similar increase in the detection efficiency with higher flow
rate as in Kangasluoma et al. (2015a) was not observed, which can be due to the differences in the
saturators between the 3010 and 3772: 3010 has a single hole reservoir type saturator while 3772 has
8 hole multitube saturator which possibly saturates the sample flow better than the one hole saturator
at higher flow rates.
3.4 Concentration calibration
As with all CPCs, the peak supersaturation, and hence the lowest detectable particle size is can be
affected by the presence of other particles in the flow, due to a combination of condensational heat
release and vapor depletion. These effects for the original WCPCs were explored by Lewis and Hering
(2013), and is evaluated here for the v-WCPC. Figure 12 shows the concentration dependent response
at four particle sizes for the v-WCPC. The maximum concentration at each size was determined by the
maximum concentration we were able to pass through the DMA. The data are corrected for dead
time, as described by Hering et al. (2005), and as is standard for most of the commercial CPCs. This
approach uses the instrument dead time multiplied by a dead time correction factor, which accounts
for the increase in effective dead time due to overlapping tails in pulses below the threshold. For this
data set the dead time correction factor was set to 1.23 to yield a linear response to concentration at
4.4 nm. Then this same dead time correction factor was applied to measurements at other sizes. The
curves of the three smallest particle sizes have a negative slope due to the reduction in
supersaturation at high concentrations caused by condensational heating (Lewis and Hering, 2013).



However, the effect is relatively small, with the detection at 1.8 nm dropping from 36% at a
concentration of 3000cm$^{-3}$ to 33% at a concentration of 90000cm$^{-3}$.
3.5 Atmospheric sampling
A fraction of the data measured from atmospheric aerosol is presented in Figure 13. The measurement
location is above a bus stop, which several busses pass daily through. The bus stop times are marked
to the figure. The background aerosol concentration during that morning was around 3 000 – 10 000
cm$^{-3}$. Clear spikes up to 200 000 cm$^{-3}$ in the measured concentrations are observed throughout the
morning, of which timing match quite well the scheduled bus departure times. From the data of Figure
13, a correlation plot between the v-WCPC and 3777 is presented in Figure 14 for concentrations
below 50 000 cm$^{-3}$. With R$^2$ of 0.99 the two CPCs show remarkably good agreement with slope of 1.02
and offset of 340 up to concentrations of 50 000 cm$^{-3}$.
4 Conclusions
Three new sub 3 nm CPCs, boosted 3010 type CPC, ADI versatile water CPC and the TSI 3777 nano
enhancer were characterized for the d50 diameter. The boosted 3010 type CPC was shown to be able
to detect tungsten oxide particles smaller than 3 nm. The vWCPC and 3777 were characterized with
similar test aerosols with two different settings: low dT settings set so that the CPCs did not detect
any ions from a radioactive charger, and high dT settings set either so that the supersaturation was at
the onset of homogeneous droplet formation (3777) or set to the largest value that avoids freezing or
boiling (v-WCPC). The d50 diameters for tungsten oxide were found to range from 1.7 nm to 2.4 nm
at low dT and from 1.4 nm to 2.2 nm at high dT for the v-WCPC. For the 3777 the d50 ranged from 1.8
nm to 2.3 nm at low dT and from 1.3 nm to 2.1 nm at high dT. Both CPCs were observed to detect
charged tungsten oxide particles better than neutral ones. The organic salt particles (TDDABr) were
detected less efficiently, with low dT d50 diameters of 3.3 nm for the v-WCPC, and 2.5 nm for the TSI-
3777. When measuring the same atmospheric aerosol the two CPCs showed a very good agreement
with regression slope of 1.02 and R$^2$ of 0.99.
From the results we can make the following conclusions: The TSI 3010 hardware can be tuned
to accomplish 3 nm particle detection by increasing the dT but not by increasing the inlet flow rate,
which is in line with Buzorius (2001). This is possibly due to not perfect flow saturation in the reservoir
type saturator as opposed to the multihole saturator of TSI 3772 and planar type saturator of
Airmodus A20 (Kangasluoma et al., 2015a). Due to the variations in the d50 with composition for the
vWCPC and 3777, their use as a detector downstream of a DMA is suggested only if the particle
composition is known and CPC calibration is made accordingly with the same particle composition, or
with sizes above the highest d50 (approximately 2.5 – 3 nm) if the particle composition is completely
unknown. The effect of particle charge on the d50 was show to be up to approximately 0.5 nm, which
has implications on to system characterizations where the fraction of charged particles can be
expected to be high (Wang et al., 2017), or CPC calibration is conducted with charged particles and
sampled particles are neutral, and high precision d50 is required.
Acknowledgements
The authors acknowledge TSI Inc. who provided the control boards and optics for the v-WCPC, and
who loaned the TSI 3777 for these experiments. The research was partly funded by European Research
Council (ATMNUCLE, 227463), Academy of Finland (Center of Excellence Program projects 1118615
and 139656), European Commission seventh Framework program (ACTRIS2 contract no 654109, PPP
and EUROCHAMP-2020), Labex ClerVolc contribution n° 228 and Maj and Tor Nessling Foundation.

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



Figure 1. TSI 3777 nano enhancer (courtesy of TSI Inc.)

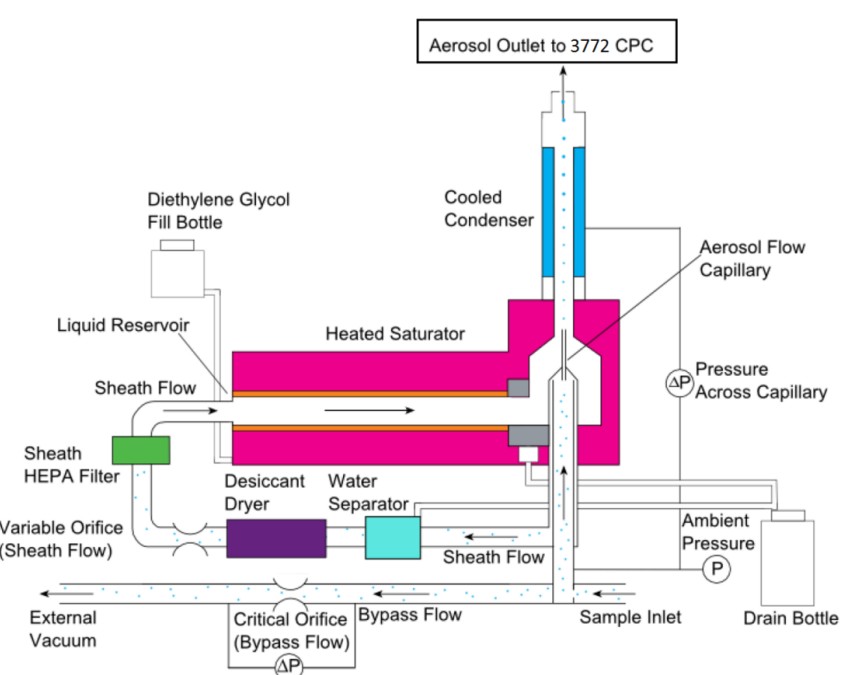

Figure 2. ADI v-WCPC





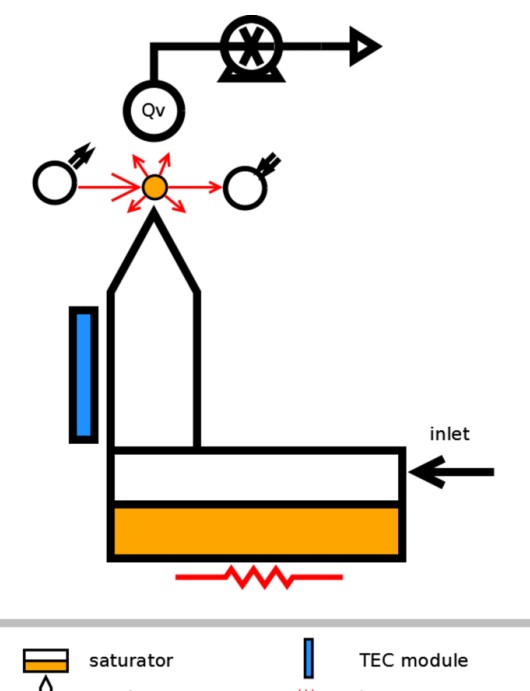

Figure 3. B3010.

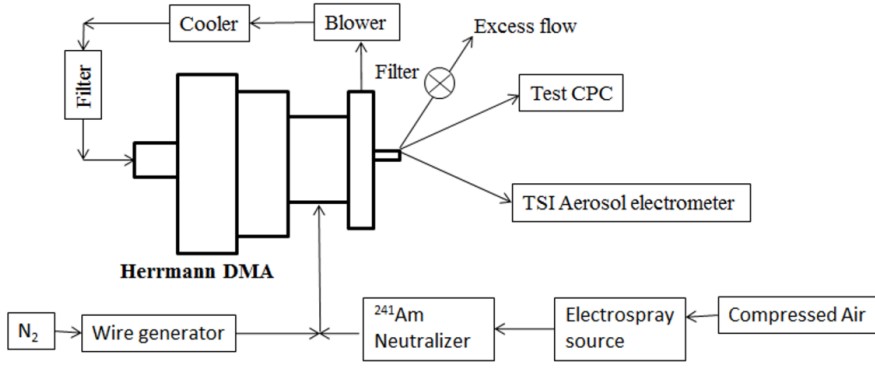

Figure 4. Experimental setup to measure d50 for charged particles





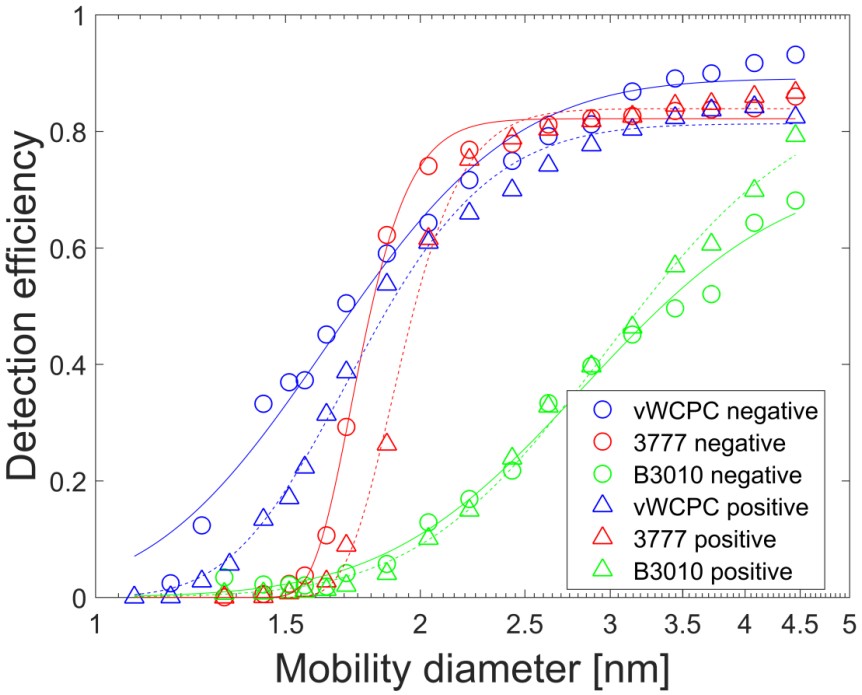

Figure 5. Detection efficiency of the CPCs as a function of size for negatively and positively charged
tungsten oxide particles at low dT settings.

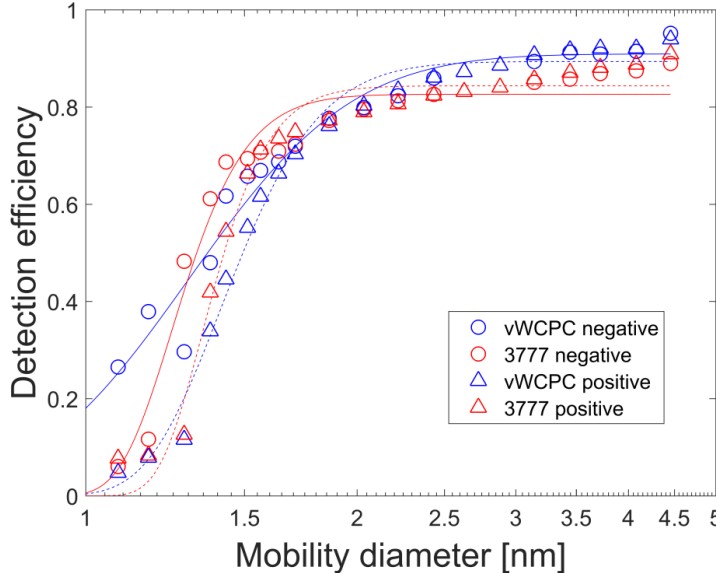






Figure 6. Detection efficiency of the CPCs as a function of size for positively and negatively charged
tungsten oxide particles at high dT settings.

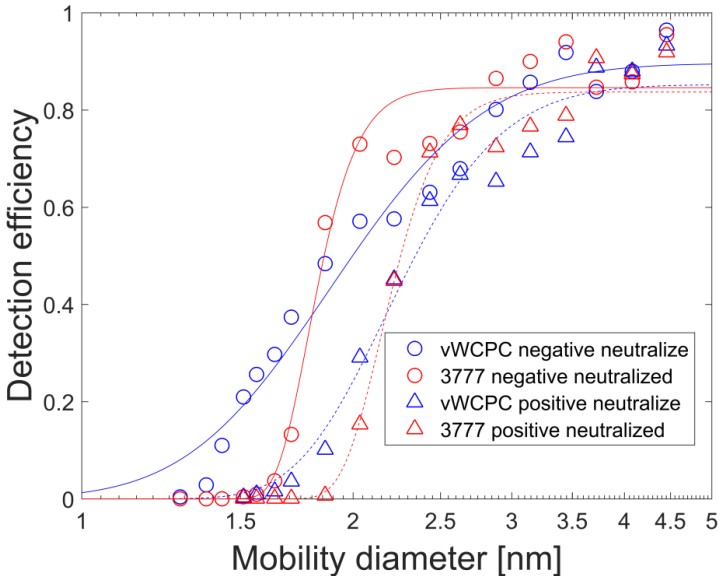

Figure 7. Detection efficiency of the CPCs as a function of size for negatively and positively charged
tungsten oxide particles that are neutralized at low dT settings.

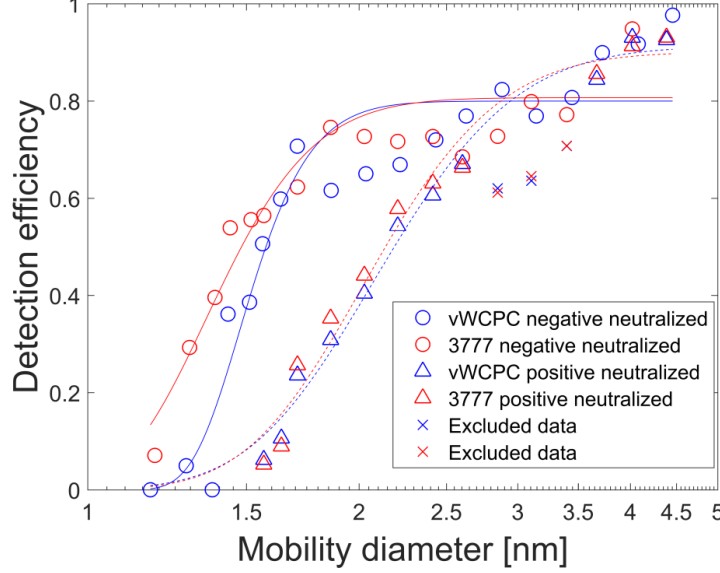

Figure 8. Detection efficiency of the CPCs as a function of size for negatively and positively charged
tungsten oxide particles that are neutralized at high dT settings.





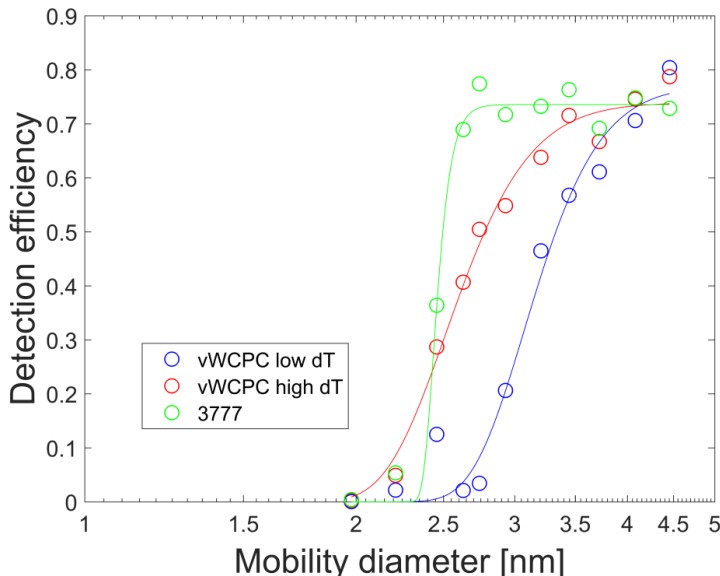

Figure 9. Detection efficiency of the CPCs as a function of size for positively charged TDDABr particles
at low and high dT settings for v-WCPC and at low dT settings for the 3777.

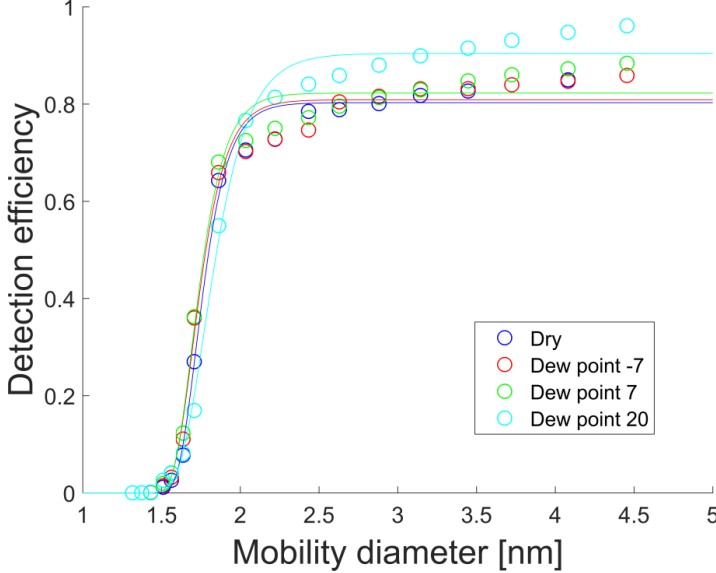

Figure 10. Detection efficiency of the 3777 as a function of the diameter and sample flow relative
humidity.



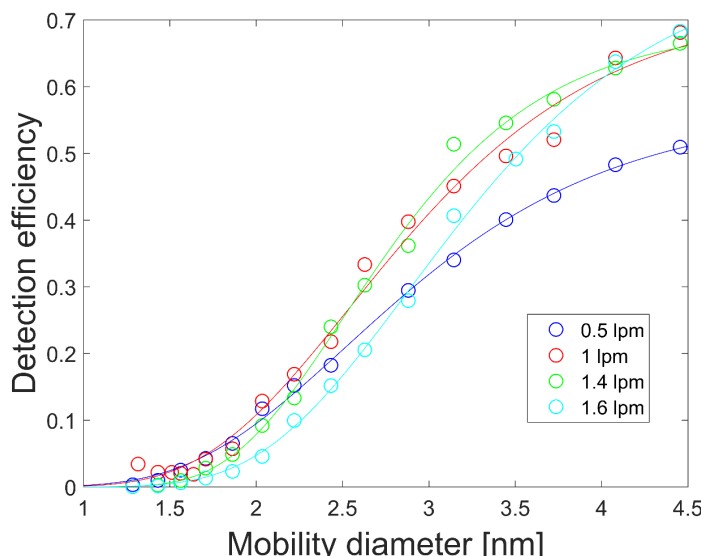

Figure 11. Detection efficiency of the B3010 as a function of the inlet flow rate.

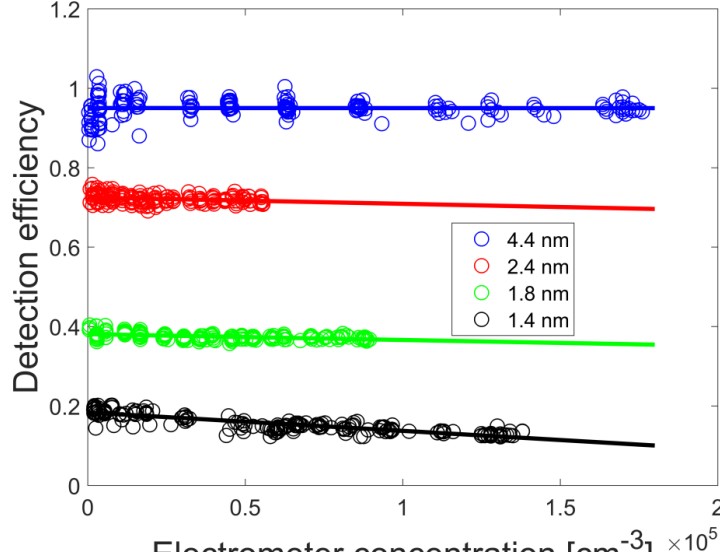

Figure 12. Ratio of the v-WCPC to the electrometer as function of the particle concentration.



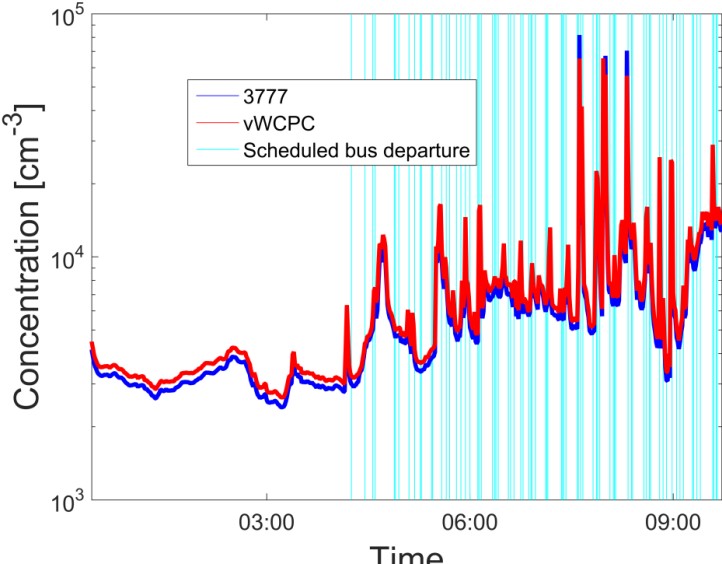

Figure 13. Concentration measured by the 3777 and v-WCPC from urban atmospheric air.

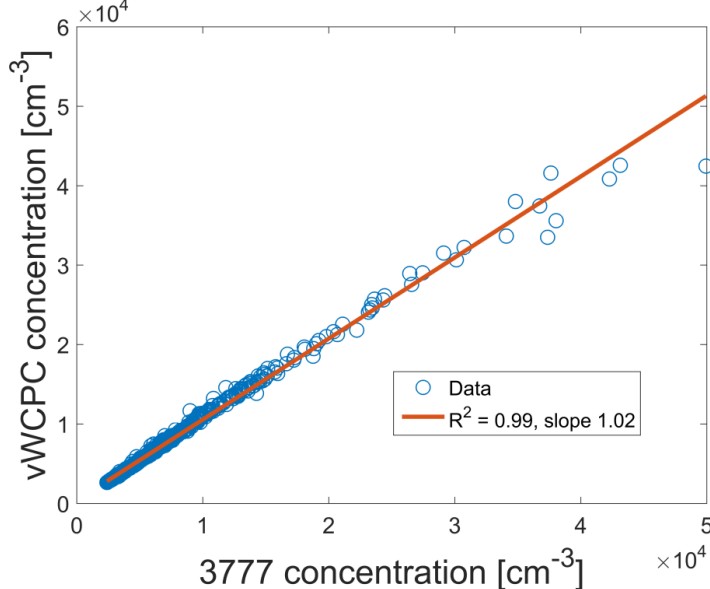

Figure 14. Correlation of the concentrations below 50 000 cm⁻³ measured by the CPCs for the same
data as in Figure 13.






Table 1. Instrument operation conditions

| Instrument | Qinlet [lpm] | Qaerosol [lpm] | Settings | Ts [oC] | Tc [oC] | Tm [oC] | To [oC] |
|---|---|---|---|---|---|---|---|
| B3010 | 1 | 1 | Low dT | 55 | 10 | | 56 |
| vWCPC | 2.2 | 0.3 | Low dT | 8 | 90 | 22 | 40 |
| vWCPC | 2.2 | 0.3 | High dT | 1 | 95 | 22 | 40 |
| 3777 | 2.5 | 0.15 | Low dT | 62 | 12 | | |
| 3777 | 2.5 | 0.15 | High dT | 70 | 7 | | |

Table 2. Indicated Cutpoints

| Conditions | Aerosol | Charging state | ADI v-WCPC | TSI-3777 | B3010 |
|---|---|---|---|---|---|
| High dT | WOx | negative | 1.4 | 1.3 | NA |
| High dT | WOx | positive | 1.5 | 1.4 | NA |
| High dT | WOx | neutral from - | 1.6 | 1.5 | NA |
| High dT | WOx | neutral from + | 2.2 | 2.1 | NA |
| Low dT | WOx | negative | 1.7 | 1.8 | 3.4 |
| Low dT | WOx | positive | 1.9 | 2 | 3.2 |
| Low dT | WOx | neutral from - | 2 | 1.9 | NA |
| Low dT | WOx | neutral from + | 2.4 | 2.3 | NA |
| High dT | TDDAB | positive | 2.8 | NA | NA |
| Low dT | TDDAB | positive | 3.3 | 2.5 | NA |
