# Peer review of "Characterization of three new condensation particle counters for sub-3 nm particle detection during"

_Atmospheric Measurement Techniques, 2016_

## Referee Comment (RC1) · Anonymous Referee #1 · 27 Jan 2017

Reviewer's comments on the Characterization of three new condensation particle counters for sub-3nm particle detection: ADI versatile water CPC, TSI 3777 nano enhancer and boosted TSI 3010

General Comments

The major outcome of this work is systematic comparisons between two newest CPCs for detecting sub-3 nm particles, which are TSI-3777 and ADI v-WCPC. The reported results are valuable to the potential users of these instruments. However, readers are expected to feel that B3010 are not compared a systematically with v-WCPC and

3777. For example, authors are showing that the detection efficiencies of v-WCPC and 3777 shifts to large sizes when TDDAB are used as the test particles. Reader would expect to see whether the butanol based instruments are also affected in a similar way or not. The same comments apply to the effect of moisture content on the detection efficiencies. Some readers may wonder whether the detection efficiencies of butanol instruments are different between unsheathed and sheathed. It is interesting to know the effect of flowrates on the detection efficiencies of B3010. However, since authors believe that the results are less impressive than their previous results obtained using 3772, it is probably not worth to presenting this results in this paper. I personally recommend to remove the results associated with B3010 in this paper, and gather and emphasize positive aspects of B3010 in another paper.

Specific Comments

Line 15-16. This statement communicates well but it sound too casual. I recommend to use more intellectual expressions since it is the first line of the abstract.

Line 24-26 Authors do not clearly state whether high ïĄĎT setting detect ions generated by bipolar ion source or not throughout the paper. Please clarify somewhere in the text.

Line 73-74. I believe that one great benefit of using water as working fluid is that water vapor is generally not considered as a source of contamination whereas the organic vapor are often considered as contaminates.

Line 90 "effect of electrical charge" sounds more proper.

Line 98-110 The statement in line 101-102 "possibly altering the detection efficiency...." is not needed at this point since the purpose of this paragraph to describe the structure of the 3777, not to comment on its performance. Instead, authors may emphasize that saturator of the 3777 has a meandering path in a metal block instead of a porous wick used in butanol based ultrafine-CPCs. It is cumbersome to remove DEG liquid absorbed in a porous wick since DEG is viscous and has a relatively

low vapor pressure.

Line 127 "Cheap second hand" sounds inappropriate. It give an impression that authors are looking down on older models. Is it true that simpler electronics of the older model made the modifications much easier?

The style of the schematic for B3010 are very different from those of 3777 and v-WCPC. If the data for B3010 are going to be included in this paper the style of the schematic should be similar to 3777 and v-WCPC.

Line 182-184 Uncertainty should include the particles size after neutralization since recombination between a charged particle and ions of opposite polarity may not transfer just electrical charge. The particle and ion may stick together to generate a larger particles as shown by Sipilä et al (2009).

Line 184-185 "it is the only method to measure d50 for neutral particles. . ." This statement is not entirely true since Winkler (2008, Science) measured the activation efficiencies for electrically neutral and charged particles using their expansion chamber.

Line 188: Title of the section 2.3 should be "Concentration Calibration and Atmospheric Sampling" to be consistent with the titles of 3.4 & 3.5.

Line 211 to 212 Resolution with respect to mobility diameter is about twice of the resolution with respect to the electrical mobility in free molecular regime. It seems more conventional to express uncertainty as the square root of a variance not as the full width half maximum.

Line 215-216 The statement "such as commonly used for mobility based . . .." does not seem necessary since the role of the bipolar ion source is already introduced in previous section

Line 243-254 The results shown in Figure 7 is very interesting although the measured values are being affected by several sources of uncertainties. It is recommended that authors discuss the sources of observed differences between positively and negatively

charged particles after neutralization. One potential source is the difference in their chemical compositions as already stated by authors in line 235-236. Isn't it also possible that the neutralization efficiency depend not only on particle size but also on the polarity of the bipolar ions due to the difference in their mobility?

Line 260-261, and Line 338-342 The statements in these lines are rather extreme or too demanding. It is generally true that compositions of freshly nucleated nanoparticles are partially known from other measurement techniques or previous studies. It is still very useful to be able to measured particle size distributions and their uncertainties in sub 3 nm range after size-classification although their chemical compositions are not known completely. For example, if we are to investigate the effect of conditioning on the freshly nucleated particles it is not important to know the chemical composition of the DMA-classified particles since the material dependence of the detection efficiencies cancels out between before and after the conditioning.

Line 265 It is unclear the "error" stated by the authors refer to what measured variables.

Line 274-276 I understand that authors would like to support the instruments developed by their colleagues, but this statement is somewhat irrelevant to the objective of this paper. In addition readers would not understand why replacing internal pump with MFC reduces the water content.

Line 308-317, 3.5 Atmospheric sampling There seems to be equal number of bus departure time which does not show clear spikes. I believe that most reader would see from Figure 13 are the followings. The number concentration generally high during traffic hours, and both CPCs reacts instantaneously to the occasional spikes in the number concentration. Readers would be able see the concentration dependence more clearly if the data in Figure 14 are plotted on a log-log scale. One-to-one lines needs to be shown. Plotting data on log-log scale does not stop authors stating that there is an offset.

Line 336: not perfect => imperfect?

---

## Referee Comment (RC2) · Anonymous Referee #3 · 14 Feb 2017

General comments:

This manuscript describes some calibration and comparisons performed with 3 CPCs, chosen for their abilities to measure particle sizes below 3 nm. Good performance data on CPCs is critical for interpreting their measurements. As such, the manuscript would make a contribution in this area. I have two main concerns and a number of minor suggestions.

Main concern 1: Overall the manuscript is fairly easy to follow but there are times when I feel it could benefit greatly from a careful review for English grammar and typographic

errors.

Main concern 2: As described specifically below, the paper often reads like a random collection of data from lab experiments and the reader is not told why these experiments were chosen, why some experiments were performed on one instrument and not another, etc. The authors should address this early in the manuscript to allow the reader to make better use of these observations.

Minor edits/questions/comments:

49: I believe that Brechtel began commercialization of their mixing-type CPC before 2011, so this statement should be modified to "the use of a mixing CPC for a booster" or some-such.

100: correct typo "pm"

108 and 124: state activity of the radioactive source

127: "cheap second hand" seems rather up to interpretation . . . I could argue that even used CPCs are not "cheap" and it's unclear how the fact that this CPC was "second hand" impacts its performance. Please consider rephrasing.

137: details -> detail

154: why was the line length made half that of the other CPCs "for the same reason"?

155: It is sometimes difficult to understand why the authors chose the parameter space for operating the instruments in the way they did. For example, why did the authors decide to measure just the 3777 at different dew points? Also why was the sample flow rate only changed on the B3010? Also why were both of the above two issues mentioned in the section devoted to aerosol generation? [Note: I now realize that an explanation for the dew point is provided in line 265. I suggest having such a sentence earlier in the paper so the reader better understands the experimental parameters.

175: insert "source" after radioactive?

[Figure]

184: remove "being" or replace with "currently"

243, 261: Awkward and possibly grammatically incorrect sentences ... please rephrase.

292: Similar to the criticism of line 155, why was concentration dependence calibration performed only for the v-WCPC? Could it not also be an important factor for the other CPCs?

342: show -> shown

---

## Author Comment (AC1) · 10 Apr 2017

Point by point response to the review is in the attached files.

Please also note the supplement to this comment:
http://www.atmos-meas-tech-discuss.net/amt-2016-408/amt-2016-408-AC1-supplement.zip
* * *

---

## Author Response (AR1)

We thank both reviewers for corrections and suggestions which improved the manuscript. Both reviewers shared the concern of the manuscript being a random collection of experimental results. Due to this concern we changed the manuscript structure so that there is a section for all CPCs, further comparison between 3777 and v-WCPC, and CPC specific experiments. The title was also edited to highlight the fact that the experiments were conducted during a workshop in Helsinki, when all CPC were brought to the same laboratory. Also, we language of the manuscript was improved.

Below are our point-to-point responses for all concerns of the reviewers, and below shown the modifications to the manuscript. Line numbers refer to the document with no track changes.

Reviewer 1

General Comments
The major outcome of this work is systematic comparisons between two newest CPCs for detecting sub-3 nm particles, which are TSI-3777 and ADI v-WCPC. The reported results are valuable to the potential users of these instruments. However, readers are expected to feel that B3010 are not compared a systematically with v-WCPC and 3777. For example, authors are showing that the detection efficiencies of v-WCPC and 3777 shifts to large sizes when TDDAB are used as the test particles. Reader would expect to see whether the butanol based instruments are also affected in a similar way or not. The same comments apply to the effect of moisture content on the detection efficiencies. Some readers may wonder whether the detection efficiencies of butanol instruments are different between unsheathed and sheathed. It is interesting to know the effect of flowrates on the detection efficiencies of B3010. However, since authors believe that the results are less impressive than their previous results obtained using 3772, it is probably not worth to presenting this results in this paper. I personally recommend to remove the results associated with B3010 in this paper, and gather and emphasize positive aspects of B3010 in another paper.
See general response above.

Specific Comments
Line 15-16. This statement communicates well but it sound too casual. I recommend to use more intellectual expressions since it is the first line of the abstract.
Line 15 Silence is golden. Removed the statement.

Line 24-26 Authors do not clearly state whether high dT setting detect ions generated by bipolar ion source or not throughout the paper. Please clarify somewhere in the text.
Added statement to line 136 "At the high dT settings both 3777 and v-WCPC are able to detect ions from a radioactive source."

Line 73-74. I believe that one great benefit of using water as working fluid is that water vapor is generally not considered as a source of contamination whereas the organic vapor are often considered as contaminates.
Edited the sentence at line 75 as: "The disadvantage of all the previous CPCs is the slight toxicity of the working fluids butanol and DEG, and these organic liquids can introduce contaminant molecules to vapor phase."

Line 90 "effect of electrical charge" sounds more proper.
Edited as suggested

Line 98-110 The statement in line 101-102 "possibly altering the detection efficiency: : :." is not needed at this point since the purpose of this paragraph to describe the structure of the 3777, not to comment on its performance. Instead, authors may emphasize that saturator of the 3777 has a meandering path in a metal block instead of a porous wick used in butanol based ultrafine-CPCs. It is cumbersome to remove DEG liquid absorbed in a porous wick since DEG is viscous and has a relatively low vapor pressure.

Thanks for the detail, edited as suggested to line 111.

Line 127 "Cheap second hand" sounds inappropriate. It give an impression that authors are looking down on older models. Is it true that simpler electronics of the older model made the modifications much easier? The style of the schematic for B3010 are very different from those of 3777 and v-WCPC. If the data for B3010 are going to be included in this paper the style of the schematic should be similar to 3777 and v-WCPC.

Replaced "cheap second hand" with "robust" at line 138. Figure 3 changed to a better fitting one. The point was to show that the saturator and condenser temperature decoupling can be done for the 3010, and it can be modified to detect particles of and smaller than 3 nm. In the 3772 the saturator and condenser temperature control is already decoupled.

Line 182-184 Uncertainty should include the particles size after neutralization since recombination between a charged particle and ions of opposite polarity may not transfer just electrical charge. The particle and ion may stick together to generate a larger particles as shown by Sipilä et al (2009).

While we agree with this comment, and added a statement about this uncertainty, Sipilä et al. 2009 only speculate about this problem in case of measuring detection efficiency of charger ions. Indeed to our knowledge such experiments have not been conducted where the recombination products are studied at molecular level, which is probably due to experimental difficulties in probing the neutral recombination products. Edited the statement at line 179-> as "This method yields uncertainties in the resulting d50 due to possibly size dependent neutralization efficiency, unknown neutralization mechanism (ion recombination leading to larger physical size, or charge transfer) and chemical composition of the neutralized particles, however, it is being the only method to measure d50 for neutral particles for sub-3 nm particles."

Line 184-185 "it is the only method to measure d50 for neutral particles: : :" This statement is not entirely true since Winkler (2008, Science) measured the activation efficiencies for electrically neutral and charged particles using their expansion chamber.

Added clarification after the statement "Winkler et al. (2008) used similar method to measure nucleation probabilities of electrically charged and neutral clusters, the difference being that they used bipolar neutralizer."

Line 188: Title of the section 2.3 should be "Concentration Calibration and Atmospheric Sampling" to be consistent with the titles of 3.4 & 3.5.

Edited as suggested

Line 211 to 212 Resolution with respect to mobility diameter is about twice of the resolution with respect to the electrical mobility in free molecular regime. It seems more conventional to express uncertainty as the square root of a variance not as the full width half maximum.

Thanks. This is correct, it should have been ±FWHM/2 (2.5%) which is closer to σ. Now corrected so that normal distribution was fitted to the THABr positively charged monomer spectrum, and from that square root of variance ($\sqrt{\sigma^2}$)) was taken as the uncertainty, resulting as uncertainty of ± 2%. Corrected at line 236 -> as "X-axis uncertainty can be taken from the Herrmann DMA resolution of approximately 20, which leads to relative uncertainty of ± 2%, which is the square root of variance of the normal distribution fitted to the tetraheptylammonium bromide positively charged monomer peak."

Line 215-216 The statement "such as commonly used for mobility based : : :." Does not seem necessary since the role of the bipolar ion source is already introduced in previous section

Removed as suggested

Line 243-254 The results shown in Figure 7 is very interesting although the measured values are being affected by several sources of uncertainties. It is recommended that authors discuss the sources of observed differences between positively and negatively charged particles after neutralization. One potential source is the difference in their chemical compositions as already stated by authors in line 235-236. Isn't it also possible that the neutralization efficiency depend not only on particle size but also on the polarity of the bipolar ions due to the difference in their mobility?

Added clarification to line 282 ->: "The difference in the d50s between the negatively and positively charged particles after neutralization are possibly mostly explained by their differences in the chemical composition. Positively charged particles contain more contaminant species than the negatively charged particles, which is still observed in the d50 when they both are neutralized: lower d50 for the neutralized negative particles than for the neutralized positive particles. For more details, refer to Kangasluoma et al. (2016b).".

The reviewer is correct in that we observed different neutralization efficiency for different polarities. However, it does not affect the obtained d50s since the data is normalized to 90% at the curve plateau region.

Line 260-261, and Line 338-342 The statements in these lines are rather extreme or too demanding. It is generally true that compositions of freshly nucleated nanoparticles are partially known from other measurement techniques or previous studies. It is still very useful to be able to measured particle size distributions and their uncertainties in sub 3 nm range after size-classification although their chemical compositions are not known completely. For example, if we are to investigate the effect of conditioning on the freshly nucleated particles it is not important to know the chemical composition of the DMA-classified particles since the material dependence of the detection efficiencies cancels out between before and after the conditioning.

Here we slightly disagree. First, there are not many experiments where the composition of the nucleating particles are exactly known, CLOUD experiment in Cern could be one. The only atmospheric experiment is by Sipilä et al. 2016, Nature. Second, even if the composition of the nucleating species is known, or partly known, it does not mean that the CPC detection efficiency curve is known for that experiment, since the CPC should be calibrated with the same aerosol. Almost always this is not possible, but it does not mean that it is not a large source of uncertainty for the d50. Indeed, we think that efforts should be focused on this problem to reduce the particle counting uncertainties.

Having for example a rough idea of the d50 with the accuracy of ± 0.5 nm can lead to uncertainties from basically -∞ to +∞ (form zero detected particles to all detected, the exact uncertainty depends on the sample size distribution) at the sizes close to the d50. This d50 uncertainty can lead to huge uncertainty in the measured particle concentration, if strictly looking at the measured particle concentration at a fixed mobility diameter.

We agree that the measurements should be done regardless of this uncertainty, but more effort should be put to understand the sources of uncertainty in the obtained concentrations (and parameters obtained from the concentrations, such as growth and nucleation rates).

Edited line 294-296 a little "To obtain accurate particle concentration measurements, these differences in the d50 imply that the CPCs should be calibrated with the same aerosol composition as with the real experiment is conducted."

Edited line 363-367 as: "Due to the variations in the d50 with composition for the vWCPC and 3777, a careful CPC calibration should be conducted with the same particle composition as of the sampled particles. If the composition of the sampled particles is completely unknown, the obtained particle concentrations at the size range of the d50 can have significant uncertainties."

Line 265 It is unclear the "error" stated by the authors refer to what measured variables.

Edited to line 309 as "Because water was observed to alter the d50 in the original laminar-flow DEG instruments, this question was examined for the 3777"

Line 274-276 I understand that authors would like to support the instruments developed by their colleagues, but this statement is somewhat irrelevant to the objective of this paper. In addition readers would not understand why replacing internal pump with MFC reduces the water content. Removed as suggested. As it was written, MFCs are usually fed by dried compressed air while internal pumps just draw ambient air without drying.

Line 308-317, 3.5 Atmospheric sampling There seems to be equal number of bus departure time which does not show clear spikes. I believe that most reader would see from Figure 13 are the followings. The number concentration generally high during traffic hours, and both CPCs reacts instantaneously to the occasional spikes in the number concentration. Readers would be able see the concentration dependence more clearly if the data in Figure 14 are plotted on a log-log scale. One-to-one lines needs to be shown. Plotting data on log-log scale does not stop authors stating that there is an offset. Changed fig14 to log-log scale. Edited to line 300-> as suggested "Clear spikes up to 200 000 cm$^{-3}$ in the measured concentrations are observed throughout the morning. The number concentration generally high during traffic hours, and both CPCs reacts instantaneously to the occasional spikes in the number concentration. From the data of Figure 13, a correlation plot between the v-WCPC and 3777 is presented in Figure 14 for concentrations below 50 000 cm$^{-3}$."

Line 336: not perfect => imperfect?

Corrected

Reviewer 2

General comments:
This manuscript describes some calibration and comparisons performed with 3 CPCs, chosen for their abilities to measure particle sizes below 3 nm. Good performance data on CPCs is critical for interpreting their measurements. As such, the manuscript would make a contribution in this area. I have two main concerns and a number of minor suggestions.

Main concern 1: Overall the manuscript is fairly easy to follow but there are times when I feel it could benefit greatly from a careful review for English grammar and typographic errors.

Main concern 2: As described specifically below, the paper often reads like a random collection of data from lab experiments and the reader is not told why these experiments were chosen, why some experiments were performed on one instrument and not another, etc. The authors should address this early in the manuscript to allow the reader to make better use of these observations. See general response above for both main concerns.

Minor edits/questions/comments:
49: I believe that Brechtel began commercialization of their mixing-type CPC before 2011, so this statement should be modified to "the use of a mixing CPC for a booster" or some-such. Edited as "DEG based mixing type CPC technology"

100: correct typo "pm"
Corrected and 124: state activity of the radioactive source

Added to line 133, it is 185 MBq

127: "cheap second hand" seems rather up to interpretation : : : I could argue that even used CPCs are not "cheap" and it's unclear how the fact that this CPC was "second hand" impacts its performance. Please consider rephrasing.
Removed as already suggested by the reviewer 1, replaced by "robust". The fact that it is second hand does not affect the performance, but relates to that the CPC is widely used, and we show that it can be modified for small particle detection too.

137: details -> detail
Corrected

154: why was the line length made half that of the other CPCs "for the same reason"?
This was badly formulated, corrected to line 161 -> as "The tubing lengths downstream of the DMA were selected based on the inlet flow rates so that the particle penetration through the tubes can be considered equal."

155: It is sometimes difficult to understand why the authors chose the parameter space for operating the instruments in the way they did. For example, why did the authors decide to measure just the 3777 at different dew points? Also why was the sample flow rate only changed on the B3010? Also why were both of the above two issues mentioned in the section devoted to aerosol generation? [Note: I now realize that an explanation for the dew point is provided in line 265. I suggest having such a sentence earlier in the paper so the reader better understands the experimental parameters.
This is assessed in comments to the general concerns of both reviewers

175: insert "source" after radioactive?
Corrected

184: remove "being" or replace with "currently"
Corrected

243, 261: Awkward and possibly grammatically incorrect sentences : : : please rephrase.
Edited to line 272 as "Figure 7 presents the d50 curves measured with the neutralized tungsten oxide particles for 3777 and v-WCPC.", and line 294-> as "To obtain accurate particle concentration measurements, these differences in the d50 imply that the CPCs should be calibrated with the same aerosol composition as the real experiment is conducted."

292: Similar to the criticism of line 155, why was concentration dependence calibration performed only for the v-WCPC? Could it not also be an important factor for the other CPCs?
Same answer as 155

342: show -> shown

Corrected

[revised manuscript text omitted]

---

## Author Response (AR2)

Dear Editor
Below the changes as suggested by the reviewer 1. We wish to thank the reviewer for taking care of
our mistake with the figure 11, which is now changed to log-log scale. Also to line 178 we added one
clarifying sentence on the dTs used by the CPCs.
Sincerely
Juha Kangasluoma

[revised manuscript text omitted]